

# Latent heating profiles from GOES-16 and its impacts on precipitation forecasts

Yoonjin Lee[1], Christian D. Kummerow[1,2], Milija Zupanski[2]

[1]Department of Atmospheric Science, Colorado State University, Fort Collins, Colorado, 80521, USA
[2]Cooperative Institute for Research in the Atmosphere, Fort Collins, Colorado, 80521, USA

*Correspondence to*: Yoonjin Lee (yoonjin.lee@colostate.edu)

**Abstract.** Latent heating (LH) is an important quantity in both weather forecasting and climate analysis, being the essential factor driving convective systems. Yet, inferring LH rates from our current observing systems is challenging at best. For climate studies, LH has been retrieved from the Precipitation Radar on the Tropical Rainfall Measuring Mission (TRMM) using model
simulations in the look-up table (LUT) that relates instantaneous radar profiles to corresponding heating profiles. These radars, first on TRMM and then Global Precipitation Measurement Mission (GPM), provide a continuous record of LH. However, temporal resolution is too coarse to have a significant impacts on forecast models. In operational forecast models such as High-Resolution Rapid Refresh, convection is initiated from LH derived from ground based radar. Despite the high spatial and temporal resolution of ground-based radars, one disadvantage of using these sources is that its data are only available over well
observed land areas. This study develops a method to derive LH from the Geostationary Operational Environmental Satellite-16 (GOES-16) in near-real time. Even though the visible and infrared channels on the Advanced Baseline Imager (ABI) provide mostly cloud top information, rapid changes in cloud top visible and infrared properties, when formulated as a LUT similar to those used by the TRMM and GPM radars, can equally be used to derive LH profiles for convective regions based on model simulations with a convective classification scheme and channel 14 (11.2μm) brightness temperatures. Convective regions
detected by GOES-16 are assigned LH from the LUT, and they are compared with LH from the Next Generation Weather Radar (NEXRAD) and one of the Dual-frequency Precipitation Radar (DPR) products, the Goddard Convective-Stratiform Heating (CSH). LH obtained from GOES-16 show similar magnitude with NEXRAD and CSH, and vertical distribution of LH is also very similar with CSH. One month analysis of total LH from convective clouds from GOES-16 and NEXRAD shows good correlation between the two products. Finally LH profiles from GOES-16 and NEXRAD are applied to WRF simulations for
convective initiation and their results are compared to investigate their impacts in precipitation forecasts. Results show that LH from GOES-16 have similar impacts as NEXRAD, and improves the forecast significantly.

## 1 Introduction

As the spatial resolution of numerical weather prediction models becomes finer, and even operational models are run at resolutions of a few kilometers, an effective way to assimilate observation data at this fine resolution has been sought
(Gustafsson et al., 2018). At a few kilometers resolution, convection can be resolved explicitly (Seity et al., 2011). However, if the model environment is not favorable for convection, updrafts and clouds will not develop in the right place. In order to correctly initiate convection in operational regional models where both accuracy and speed are important, observed latent heating (LH) can be added in the model in the data assimilation cycle. LH is not only important to initiate convection, it also contributes to the intensification of convection. Adding LH induces lower level convergence and upper level divergence, thereby inducing
convection, and it has become an important procedure that many operational models use for the initialization of convective events (Weygandt and Benjamin, 2007; Gustafsson et al., 2018).





The National Oceanic and Atmospheric Administration (NOAA)'s operational models, the Rapid Refresh (RAP) and High-Resolution Rapid Refresh (HRRR), both use observed latent heating to drive convection, but in different ways (Benjamin et al.,

2016). RAP uses digital-filter initialization (Peckham et al., 2016) while HRRR replaces modeled temperature tendency with the observed LH (Benjamin et al., 2016) from the Next Generation Weather Radar (NEXRAD), which is a ground-based radar network over the United States. For this operational purpose, LH data must be available continuously in near-real time. Therefore, ground-based radars which have high spatial and temporal resolutions similar to HRRR's resolution are used to calculate LH from NEXRAD reflectivity. While suitable for the HRRR region over the Contiguous United States (CONUS), the

method is not applicable to regions beyond radar coverage such as the Gulf of Mexico and even some mountainous areas.

Satellite data are used to infer climatology of LH over the globe. CloudSat which carries a W-band radar that is sensitive to light precipitation but experiences attenuation with heavy precipitation is used to derive LH for shallow precipitating regions (Huaman and Schumacher, 2018). Nelson et al., 2016 and Nelson and L'Ecuyer, 2018 created an a priori database using model simulations

from the Regional Atmospheric Modeling System (RAMS) and used a Bayesian Monte Carlo algorithms to find the most appropriate LH profiles from the database for shallow convective clouds. For deeper convection, satellites that carry instruments with lower frequencies such as Tropical Rainfall Measuring Mission (TRMM) and Global Precipitation Measurement Mission (GPM) satellites are more appropriate to retrieve LH. The Precipitation Radar (PR) on TRMM was the first meteorological radar in space, designed to provide vertical distributions of precipitation over the tropics (Kummerow et al., 1998). From its three-

dimensional hydrometeor observations, vertical profiles of LH have been retrieved. There are several retrieval algorithms using PR: Goddard Convective-Stratiform heating (CSH; Tao et al., 1993), Spectral Latent Heating (SLH; Shige et al., 2004), Hydrometeor heating (HH; Yang and Smith, 1999), and Precipitation Radar Heating algorithm (PRH; Satoh and Noda, 2001). Among these algorithms, CSH and SLH are the two most widely used products. Most recent versions of monthly gridded CSH and SLH products have spatial resolution of 0.25°×0.25° and 0.5°×0.5° respectively with 80 vertical layers and have been used to

provide valuable insights on heat budgets and atmospheric dynamics over the tropics (Schumacher et al., 2004; Chan and Nigam, 2009; Zhang et al., 2010; Liu et al., 2015; Huaman and Takahashi, 2016). The CSH and SLH algorithms have improved since their first development, and both algorithms are also applied to Dual-frequency Precipitation Radar (DPR) data on GPM, the successor of TRMM, to continue the climate record of LH and expand the regions of interest to mid-latitude.

CSH and SLH both rely on a lookup table (LUT) based on cloud resolving model simulations. Inputs that are used to look for LH profiles in these LUT are different, but their common inputs to the LUT are echo top height and surface rainfall rate as well as convective-stratiform flag. Echo top height is important in determining the depth of heating in the vertical, and surface rainfall rate is a good indicator for the intensity of maximum heating. Even though the methods use different model simulations to create the LUT, and differ in other details, they seem to exhibit similar distributions when they are averaged spatially or temporally

(Tao et al., 2016).

Although these products are considered instantaneous heating, their temporal resolutions are low compared to 15-minute or hourly observations available from ground-based radars. The current generation of geostationary observing systems (e.g., GOES-R, Himawari, GEO-KOMPSAT-2) is required to achieve a comparable sampling rate to ground-based radars. The visible (VIS)

and infrared (IR) sensor on geostationary satellite, unfortunately, cannot provide as much vertical information as active sensor do in the presence of thick clouds, but their data contain cloud top information, and rapid refresh provides important information





about a cloud's convective nature. Cloud top information from geostationary data is included when creating cloud analysis during data assimilation (Benjamin et al., 2016), and thus LH retrieved based on cloud top temperature, can be useful in the forecast model by keeping consistency of retrieved LH with the updated cloud analysis.


This study examines if cloud top information from the Geostationary Operational-Environmental Satellite-16 (GOES-16) Advanced Baseline Imager (ABI), coupled with convective cloud identification can be sufficient to approximate NEXRAD-derived LH. Following the lead of spaceborne radar LH algorithms, a LUT is created using model simulations. Once convective clouds are determined by using 10 consecutive one-minute ABI data, LH profiles for convective clouds are found in the LUT

based on cloud top temperature of the convective cloud. Unlike DPR products that are not available continuously, ABI data in mesoscale sector mode are provided with one-minute interval, and thus LH can be obtained from GOES-16 as frequently as NEXRAD, making it possible for initiating convection during the forecast. LH from GOES-16 can be beneficial over the regions without radar coverage such as ocean or mountainous regions where beam blockage by terrain degrades the quality of radar data.

Detailed descriptions of CSH and SLH products from GPM satellite and how NEXRAD converts reflectivity to LH are provided, followed by the retrieval process using GOES-16 ABI. One case study is provided to compare vertical profiles of LH from GOES-16 with other radar products, and statistical results using one-month of data are provided to evaluate whether total convective heating rates from GOES-16 are comparable to the ones from NEXRAD. Lastly, a Weather Research and Forecasting (WRF) simulation using LH from GOES-16 and NEXRAD is presented to compare impacts of LH from the two datasets in

convective initialization.

## 2 Existing LH retrieval methods

### 2.1 Radiosonde networks

LH is not an easily measurable quantity as it is almost impossible to single out temperature changes by phase changes from the total observed temperature changes. However, heat and moisture budget studies have been conducted using sounding network in

a field campaigns, and apparent heat sources ($Q_1$) and apparent moisture sinks ($Q_2$) from the budget study can be expressed as a function of LH (Yanai et al., 1973; Johnson 1984; Demott 1996). It is achieved using a diagnostic heat budget method which is first presented by Yanai et al. 1973 (Tao et al., 2006). Over a certain horizontal area, $Q_1$ can be expressed through the equation below that includes LH (Tao et al., 2006).

$$Q_1 - Q_R = \bar{\pi}\left[-\frac{1}{\bar{\rho}}\left(\overline{\frac{\partial \bar{\rho}w'\theta}{\partial z}}\right) - \overline{\nabla \cdot V'\theta'}\right] + \frac{1}{c_p}\left[L_v(c-e) + L_f(f-m) + L_s(d-s)\right] \qquad (1)$$

where prime denotes deviations from horizontal averages, which is denoted by upper bar. $Q_R$ is the radiative heating rate, $\theta$ is potential temperature, $\pi$ is non-dimensional pressure, $\rho$ is air density, $c_p$ is specific heat at constant pressure and R is gas constant for dry air. $L_v$, $L_f$, and $L_s$ represent the latent heats of condensation, freezing, and sublimation while c, e, f, m, d, and s represent each microphysical process of condensation, evaporation, freezing, melting, deposition, and sublimation, respectively. The last six terms on the right-hand side that include these microphysical processes are LH from phase changes. Since $Q_1$ can be obtained

using vertical profiles of temperature, moisture, and wind data observed during the field campaign (Tao et al., 2006), the observed $Q_1$ is used to indirectly validate GPM LH products that are retrieved together with $Q_1$.



### 2.2 CSH and SLH from GPM DPR

DPR has two operational LH algorithms: CSH and SLH. In the GPM products, LH is provided along with additional variables: $Q_1$-$Q_R$ and $Q_2$ in SLH and $Q_1$-$Q_R$-LH, $Q_R$, and $Q_2$ in CSH as well as the rain type (Tao et al., 2019). These algorithms were first

developed for TRMM data, but have been adapted to GPM data. Both algorithms use cloud resolving model simulations to create a LUT relating hydrometeor profiles to modeled heating rates. Although there is no direct measurement for LH to validate the results, retrieved $Q_1$ and $Q_2$ are compared instead with sounding data from various field campaigns through the method mentioned in section 2.1. The evolution of these products is well summarized in (Levizzani et al., 2020), but each algorithm is briefly explained here.


The CSH algorithm was first introduced by Tao et al. 1993. The initial algorithm by Tao et al.1993 used surface rainfall rate and amount of stratiform rain as inputs to the LUT, but the LUT has been improved by increasing the number of LH profiles, using finer resolution in simulations, and adding new inputs such as echo-top heights and low-level vertical reflectivity gradients (Tao et al., 2019). For high-latitude regions observed by the GPM satellite, new LUTs have been created with simulations from NASA

Unified-Weather Research and Forecasting model which is known to be suitable for high latitude weather system (Levizzani et al., 2020). Inputs to this new LUT are surface rainfall rate, maximum reflectivity height, freezing level height, echo top height, decreasing flag (whether reflectivity values drop by more than 10dBZ toward the surface or not), and maximum reflectivity intensity (Tao et al., 2019).

The SLH algorithm is based on Shige et al. 2004 and Shige et al. 2007. For tropical regions, the LUT is created for three different rain types; convective, shallow stratiform, and anvil (or deep stratiform) clouds. Inputs to the LUT are precipitation top height (PTH), precipitation rate at the surface ($P_s$), precipitation rate at the level that separates upper-level heating and lower-level heating ($P_f$) and precipitation at the melting level ($P_m$). Once non-convective rain is separated into either shallow stratiform or anvil, a vertical profile for anvil cloud is chosen based on $P_m$, and magnitudes of upper level heating and lower level cooling

are normalized by $P_m$ and ($P_m$ - $P_s$), respectively. For convective and shallow stratiform clouds, a vertical profile corresponding to the PTH is chosen, and then upper-level heating and lower-level heating are normalized by $P_f$ and $P_s$, respectively. For DPR, a new LUT is created for mid and higher latitude to account for expanded latitudinal coverage by GPM. For higher latitude regions, six precipitation types (convective, shallow stratiform, three types of deep stratiform, and other) instead of three are used, and therefore six respective LUTs exist. Inputs to these LUTs are precipitation type, PTH, precipitation bottom height,

maximum precipitation, and $P_s$.

Figure 1 shows monthly gridded products from these two algorithms over CONUS for July of 2020 at three different heights as well as their vertically integrated heating rates. Overall horizontal patterns in the two products look similar, but there is a difference in the vertical. At 2km or 5km, CSH tends to show higher heating rate especially over the mid-latitude, while at 10km,

SLH shows higher heating rates. In addition, SLH tends to have larger cooling rates throughout the layers. If integrated over the whole vertical layers, CSH tends to show higher heating rates in general. These discrepancies would be attributed to different configuration setup such as microphysical scheme used to run simulations for the LUT. The results demonstrate that the vertical profiles of LH are highly dependent on the simulations that comprise the LUT as well as different inputs to the LUTs.




**Figure 1: Monthly gridded LH from CSH at (a) 2km, (c) 5km, (e) 10km, and (g) vertically integrated LH from CSH and LH from SLH at (b) 2km, (d) 5km, (f) 10km, and (h) vertically integrated LH from SLH.**




Orbital data for these products have finer spatial resolution of 5km, and although results may be interpreted as "instantaneous" LH, the temporal resolution is too coarse to have much impacts on regional forecast models that are initialized hourly if not more frequently. These scales are consistent with ground-based radar data which is why LH derived from ground-based radar is used almost universally.

**2.3 LH from NEXRAD**

In the operational HRRR model, LH profiles retrieved using radar reflectivity replace modeled LH profiles so that appropriate heating rate can help initiate convection. LH profiles in this case are obtained through a simple empirical formula that converts radar reflectivity to LH. In Eq. (2), reflectivity is converted to potential temperature tendency using model pressure field. This equation is only applied when radar reflectivity exceeds 28dBZ. The threshold of 28dBZ was chosen based on the effectiveness

of adding heating from reflectivity in HRRR (Bytheway et al., 2017).

$$T_{ten} = \frac{1000^{R_d/c_{pd}}}{p} \frac{(L_v+L_f)Q_s}{n \cdot c_{pd}} \quad \text{where } Q_s = 1.5 \times \frac{10^{z/17.8}}{264083} \tag{2}$$

       z: grid radar/lightning-proxy reflectivity

       $T_{ten}$: temperature tendency

       p: background pressure (hPa)

205         $R_d$: specific gas constant for dry air

       $c_{pd}$: specific heat of dry air at constant pressure

       $L_v$: latent heat of vaporization at 0°C

       $L_f$: latent heat of fusion at 0°C

       n: number of forward integration steps of digital filter initialization


$T_{ten}$ in Eq. (2) is produced in K/s to meet the needs during the short-term forecast. Although heating rate is not a general output in the forecast model, it is calculated every time step by dividing temperature change from microphysical scheme by time step which is usually on the order of few tens of seconds. Therefore, this empirical formula is developed to produce LH consistent with the model framework so that LH added does not produce computational instability when ingested.

**3 LH profiles from GOES-16**

The current operational geostationary satellite, GOES-16, carries the Advanced Baseline Imager (ABI), an instrument with 16 VIS and IR channels. Mesoscale sectors, which are manually moved around to observe interesting weather events, provide data in one-minute intervals. Such high temporal resolution data have helped observe cloud developments in more detail. Using this high temporal resolution ABI data, convective clouds are detected, and LH profiles for the detected clouds are assigned from a

LUT. The LUT is created running the Weather Research and Forecasting (WRF) model simulations. While CSH and SLH algorithm look for LH profiles in a model-based LUT according to precipitation type and precipitation top height, the LUT for GOES-16 ABI is created for convective clouds that appear bright and bubbling from ABI according to brightness temperature ($T_b$) at channel 14 (11.2μm), which is a good indicator of cloud top temperature. LH is not assigned for stratiform clouds from GOES-16 as LH from stratiform clouds are not usually used to initiate convection in the forecast model. Once convective clouds

are detected using temporal changes in reflectance and $T_b$, LH profile corresponding to the $T_b$ of the detected cloud is assigned from the LUT.





### 3.1 Definition of convection in model simulations and GOES-16 ABI

In order to make a LUT for LH profiles of convective clouds, convective grid points need to be defined in the model simulation. Convection can be defined in several different ways depending on variables that are available, but the most direct and accurate

way of defining it would be to use vertical velocity (Zipser and Lutz, 1994; LeMone and Zipser, 1980; Xu and Randall, 2001; Houze 1997; Steiner et al., 1995; Del genio et al., 2012; Wu et al., 2009). Steiner et al., 1995 and Houze 1997 suggested that convective regions tend to have vertical velocity greater than 1 ms$^{-1}$, and many previous studies that used vertical velocity to define convection used a threshold of 1 ms$^{-1}$ (LeMone and Zipser, 1980; Xu and Randall, 2001; Wu et al., 2009). Similarly, this study uses a vertical velocity threshold to define the convective core as it is one of prognostic variables in the model simulations.

However, in this study, a vertical velocity threshold is defined at a layer that has maximum hydrometeor contents. This is intended to exclude potentially high values of negative vertical velocity that can occur at high levels in the cloud if evaporative cooling is present.

The vertical velocity threshold is chosen by comparing fractions of convective regions from using different thresholds with

observed convective fractions from using GOES-16 convection detection algorithm so that it best represents convective area observed from GOES-16. This study uses a convection detecting algorithms for GOES-16 ABI from Lee et al. 2021. It uses mesoscale sector data with one-minute interval to detect convective regions from ABI imagery. Two separate detection methods are proposed for vertically growing clouds in early stages and mature convective clouds that move rather horizontally once they reach the tropopause and often have overshooting tops. A detailed description of the methods can be found in Lee et al. 2021, but

it is briefly explained here. The method for vertically growing clouds measures $T_b$ decrease over 10 minutes for two water vapor channels, and if the decrease is greater than the designated threshold (-0.5K/min for channel 8 and -1.0K/min for channel 10), it assigns the pixel as convective. For mature convective clouds, the method looks for grid points that have continuously high reflectance (reflectance greater than 0.8), low $T_b$ ($T_b$ less than 250K), and lumpy cloud top (horizontal gradient values between 0.4 and 0.9) over 10 minutes. Lumpiness of the cloud top is calculated using the Sobel operator, which is commonly used for

edge detection. These thresholds are chosen based on one-month analysis against "PrecipFlag" from the Multi-Radar/Multi-Sensor System (MRMS), which classifies precipitation types combining data from ground-based radar and rain gauge observations. Combining the two methods yielded false alarm rates of 14.4% and a probability of detection of 45.3% against the ground-based radar product, but 96.4% of the false alarm cases were at least raining. Combining the two methods provides results comparable to radar product, and these methods are rather simple and fast. These methods detect any type of convective

region, and therefore, the analysis is conducted without distinguishing different types of convective clouds.

Table 1 shows convective fractions using the GOES-16 convection detecting algorithm and using different vertical velocity thresholds in the model outputs. Using higher thresholds can eliminate non-convective grids, but at the same time, it will only include the strongest part of convective regions. Using 1.5m/s shows a fractional area closest to the observed fraction, and

therefore, 1.5m/s is used to define convection in the model output. This number is actually similar to values used in some previous modeling studies (1m/s in LeMone and Zipser 1980, Xu and Randall 2001, and Wu et al., 2009) and a satellite-based study (2-4m/s in Luo et al., 2014).

**Table 1.** Fraction of convective area in observation and using different vertical velocity thresholds in the model output.

| Observation | 1m/s | 1.5m/s | 2m/s | 3m/s | 4m/s |
|---|---|---|---|---|---|
| 1.34% | 1.86% | 1.19% | 0.86% | 0.52% | 0.34% |




### 3.2 Model simulations used to create a lookup table

11 convective cases are simulated using WRF to obtain enough samples to populate each cloud top temperature bin. The convective cases are chosen over CONUS within NEXRAD network during May to August in 2017 or 2018. All simulations use the same configuration in Table 2 to avoid discrepancy between simulation results. Brightness temperatures ($T_b$s) at 11.2μm are

calculated using the Community Radiative Transfer Model (CRTM). In each scene, convective grid points are defined by the threshold found in the previous section (1.5m/s), and LH profiles from the convective grid points with the same channel 14 $T_b$ are averaged to produce mean profiles for each $T_b$ bin of the LUT. LH profiles gathered in the LUT are provided in K/s as for NEXRAD.

**Table 2.** Table for WRF simulation setup.

| Version | WRFv3.9 |
|---|---|
| Spatial resolution | 3km |
| Time step | 10 seconds |
| Microphysical scheme | Aerosol-aware Thompson scheme (The original scheme is modified to produce vertical profiles of LH as outputs) |
| Planetary boundary layer | Mellor-Yamada Nakanishi Niino (MYNN) Level 2.5 and Level 3 schemes |
| Land surface model | Rapid update cycle (RUC) land surface model |
| Long wave and short wave radiation physics | Rapid radiative transfer model for general circulation models (RRTMG) schemes |

### 3.3 Mean LH profiles according to cloud top temperature

LH profiles of convective clouds from 11 WRF simulations are collected according to 16 bins of the minimum cloud top temperature at 11.2μm. The sixteen bins range from below 200K to above 270K with a bin size of 5K. Figure 2 shows mean

vertical profiles of LH in each bin. All profiles exhibit slightly negative LH near the ground due to evaporation, but positive LH is shown at most layers. It is also nicely shown in the figure that as the $T_b$ decreases, the profile stretches up in the vertical. Interestingly though, the maximum heating rate is not perfectly proportional to $T_b$. Considering the maximum LH that is allowed in HRRR model, which is 0.01K/s, these values seem quite reasonable. Table 3 shows mean surface precipitation rate for each bin. Precipitation rate is inversely proportional to $T_b$ in Table 3. This is expected as deeper and higher clouds tend to precipitate

more. This provides more evidence that mean LH profiles for each bin can reasonably be obtained from GOES-16.







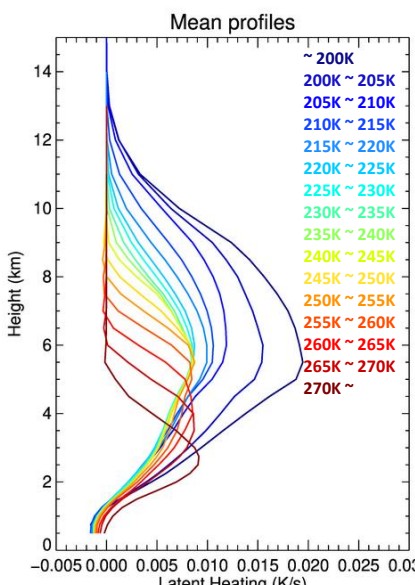

**Figure 2: Mean vertical profiles for each cloud top temperature bin.**


**Table 3.** Table of mean precipitation rate for each cloud top temperature bin.

|  | Mean precipitation rate (mm/hour) |
|---|---|
| ~200K | 48.3 |
| 200K ~ 205K | 42.9 |
| 205K ~ 210K | 42.1 |
| 210K ~ 215K | 37.9 |
| 215K ~ 220K | 33.6 |
| 220K ~ 225K | 27.7 |
| 225K ~ 230K | 21.8 |
| 230K ~ 235K | 18.8 |
| 235K ~ 240K | 16.8 |
| 240K ~ 245K | 16.4 |
| 245K ~ 250K | 14.0 |
| 250K ~ 255K | 13.2 |
| 255K ~ 260K | 11.0 |
| 260K ~ 265K | 9.2 |
| 265K ~ 270K | 6.9 |
| 270K ~ | 4.7 |





## 4 Comparisons of LH profiles between GPR DPR, NEXRAD, and GOES-16 ABI

4.1 A case study on 18 June 2019

LH from three different instruments, GOES-16 ABI, NEXRAD, and GPM DPR are examined for comparison. Methods using GOES-16 and DPR products are similar in the sense that they use cloud top height or PTH to look for mean profiles in the LUT created with model simulations, although DPR has additional parameters such as surface rain rate which is used to vary the magnitude of the heating rate. In contrast, NEXRAD uses an empirical formula to convert radar reflectivity to LH regardless of

PTH. They are all instantaneous heating, but provided in different units. LH from GOES-16 and NEXRAD are in K/s to easily match with modeled heating rate, while DPR products are in K/hour. Therefore, LH in K/hour from DPR products are converted to K/s for comparison.

A scene on 18 June 2019 is shown in Fig. 3 to compare how each product determines precipitation type (convective or

stratiform) which is one of the major factors in estimating LH profiles. The regions with reflectivity greater than 28dBZ in Fig. 3a are regions where LH is estimated from NEXRAD reflectivity to be used in HRRR, but not necessarily convective regions. These regions are larger than convective regions defined by DPR products in Fig. 3c and include some of the stratiform regions assigned by DPR. Pink regions on top of the visible image at channel 2 (0.65μm) in Fig. 3b are convective regions detected by GOES-16, and represent the smallest regions compared to others. Even though areal coverage differs by the methods, locations

of convective core matches well between the products.

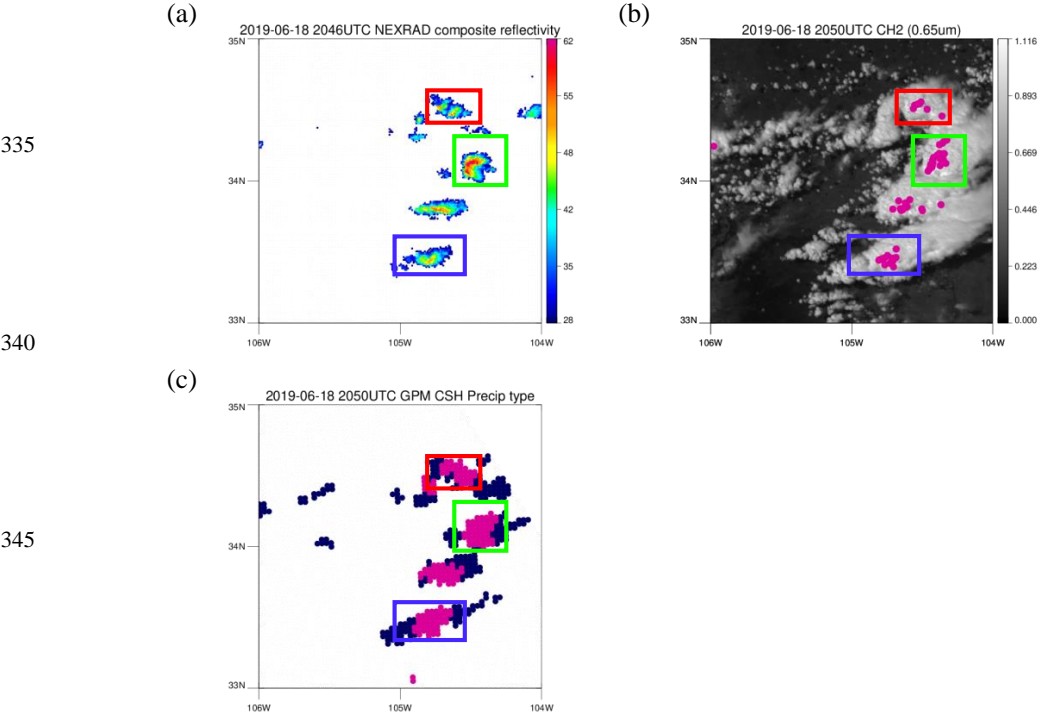

**Figure 3: A scene on 18 June 2019. (a) NEXRAD composite reflectivity. Only the regions with reflectivity greater than 28dBZ are shown in colors. Color bar is in dBZ. (b) Convective regions detected by GOES-16 are colored in pink on top of GOES-16 visible image at channel 2 (0.65μm). (c) Precipitation type defined by CSH. Convective regions are colored in pink while stratiform regions are colored in navy.**





Clouds in the colored boxes in Fig. 3 are all convective clouds, but in different evolutional stages. Clouds in red, green, and blue boxes respectively have high, low, and mid-level cloud top temperature. LH profiles from NEXRAD, GOES-16, and CSH for these clouds are interpolated into the same WRF grid with 3km resolution for comparison in Figs. 4, 5, and 6. CSH provides LH for both convective and stratiform regions, and thus different colors of lines in Figs. 4c, 5c, and 6c represent different cloud type.

Lines with light blue color are LH profiles of convective grid points in the red box, and while the blue line is the mean of these profiles. Similarly, LH profiles of each stratiform gird point are in light green, while the mean of these profiles is in dark green. The total mean LH profile is colored in red. Convective LH profiles from CSH shows heating throughout the vertical layers as expected, except near the surface due to evaporation at lower levels. LH profiles in stratiform regions show cooling at low levels below a melting level and heating above. LH profiles from GOES-16 (GOES LH) corresponding to the three convective clouds

are shown in Figs. 4b, 5b, and 6b, light blue line being each profile and blue line representing the mean. Even though mean profiles are assigned from GOES-16 for each convective cloud, a number of different lines are shown in the figure due to spatial interpolation. When GOES LH and CSH are compared, the mean profile of convective LH from CSH in blue (Figs. 4c, 5c, and 6c) is similar to GOES LH in blue (Figs. 4b, 5b, and 6b) both in terms of the magnitude and the vertical shape.

On the other hand, LH from NEXRAD (NEXRAD LH) shows a different vertical profile than GOES LH or CSH which uses the LUT consisting of model simulations. GOES LH or CSH peak around the middle of the atmosphere while NEXRAD LH in convective core (Figs. 4a, 5a, and 6a) tends to peak at low levels where radar reflectivity is high. At low levels where model simulations have cooling, NEXRAD LH does not show cooling due to Eq. (2) which is designed to only produce positive values. This heating at lower levels can help increase buoyancy in lower atmosphere, and thus, convection can be effectively initiated

from the added heating.

Although their vertical shape is different, the magnitude of the NEXRAD LH is similar to the other products. Overall values of mean LH profile from NEXRAD in blue are slightly smaller than mean profile of GOES LH or mean convective LH profile from CSH (blue line), but are closer to the total mean profile of CSH (red line), which indicates that 28dBZ threshold might include

some stratiform regions as well. A smaller mean of NEXRAD LH is mainly attributed to anvil regions where reflectivity greater than 28dBZ only exist at few vertical layers and 0dBZ elsewhere.








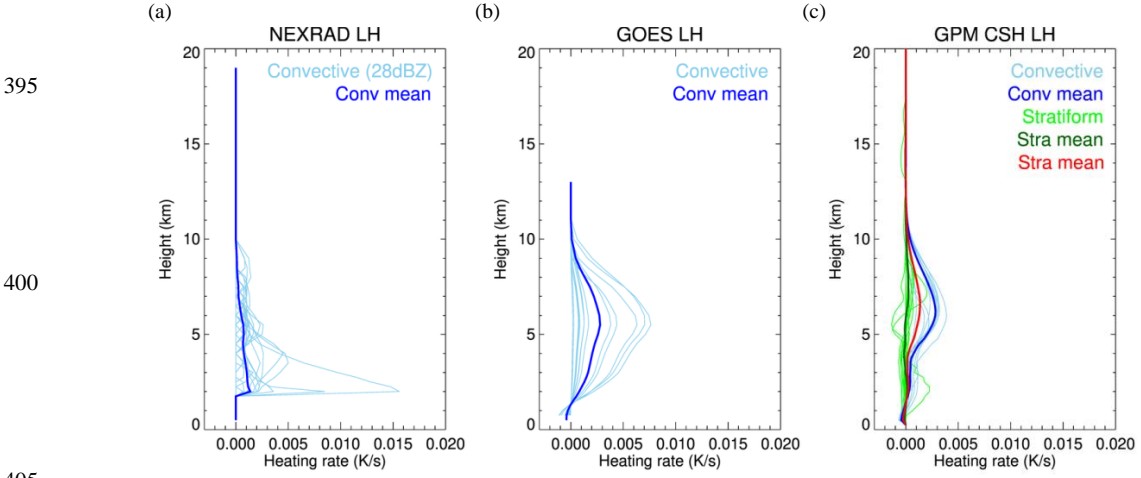

**Figure 4: LH profiles from (a) NEXRAD, (b) GOES-16, and (c) CSH for the red box region. Light blue lines are each LH profile for convective grid point and blue line is a mean profile of the light blue lines. In (c), each LH profile for stratiform grid point is coloered in light green and its mean profile is colored in dark green. The total mean of LH profiles for CSH is colored in red.**

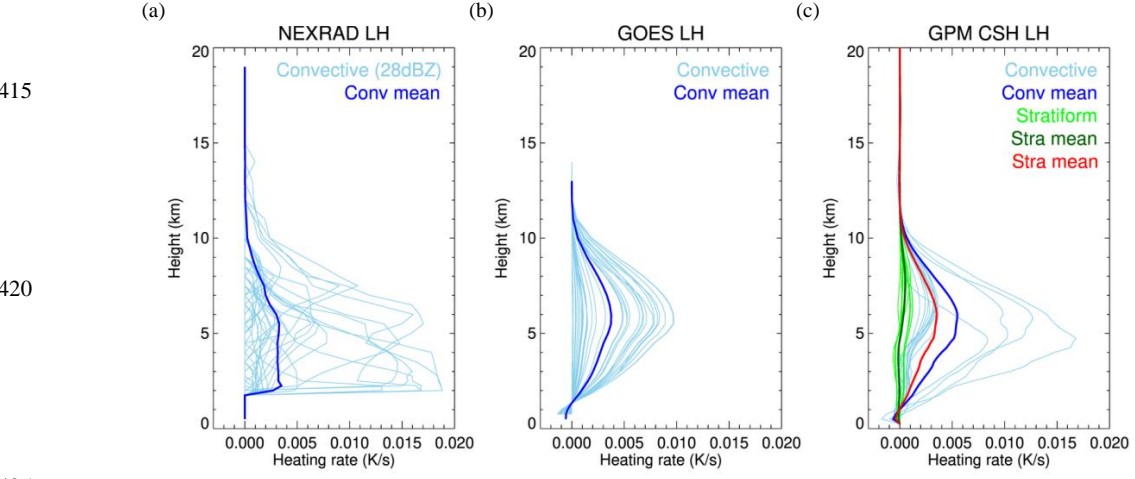

**Figure 5: Same as Fig. 4, but for the green box region.**



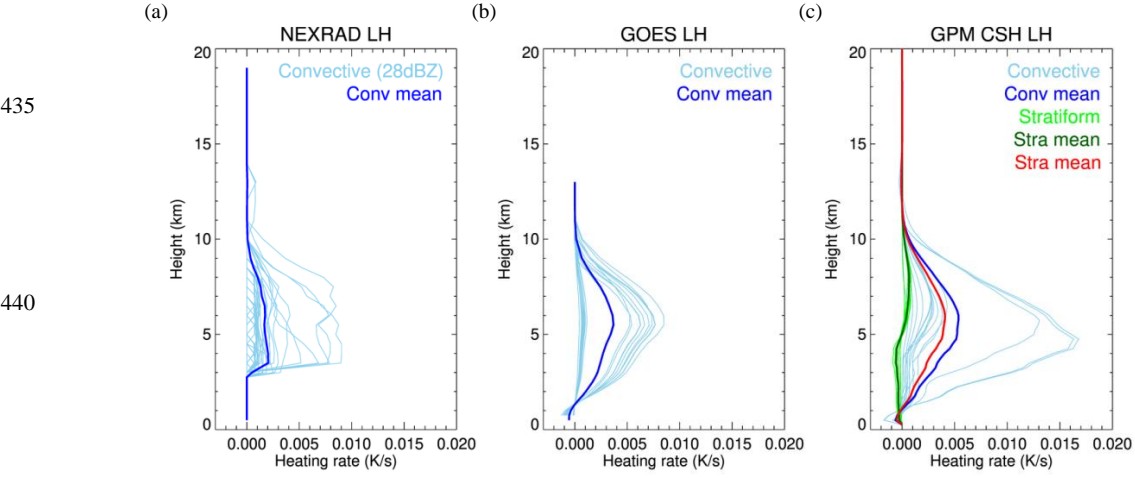

Figure 6: Same as Fig. 4, but for the blue box region.

Even though the mean NEXRAD LH is smaller, the total LH for the region can be similar when it is added up over the region

due to broader area determined by the threshold of 28dBZ in Fig. 3a than GOES-16 detection in Fig. 3b. Therefore, the total LH

of each cloud is again compared between the three products. Figure 7 shows vertical profiles of LH that are horizontally summed

over each convective cloud, each color representing colors of the three box regions. As mentioned before, the altitude that

NEXRAD LH peaks is different from the other two products. As seen from the different x-axis used in Fig. 7c, the magnitude of

total CSH LH is much larger than the other two products, and this is probably because CSH classifies broader convective regions

than GOES-16 (Fig. 3) but has higher LH values. On the other hand, the magnitude of the total heating from NEXRAD and

GOES-16 is very similar. Finally, the total LH of each region is obtained by summing up the vertical profiles in Fig. 7 and

presented in Table 4. The total LH is shown to be similar between NEXRAD and GOES-16, although GOES LH is slightly

larger. Despite the smaller mean of NEXRAD LH that was shown in Figs. 4, 5, and 6, it shows a good agreement in total heating

between GOES-16 and NEXRAD.









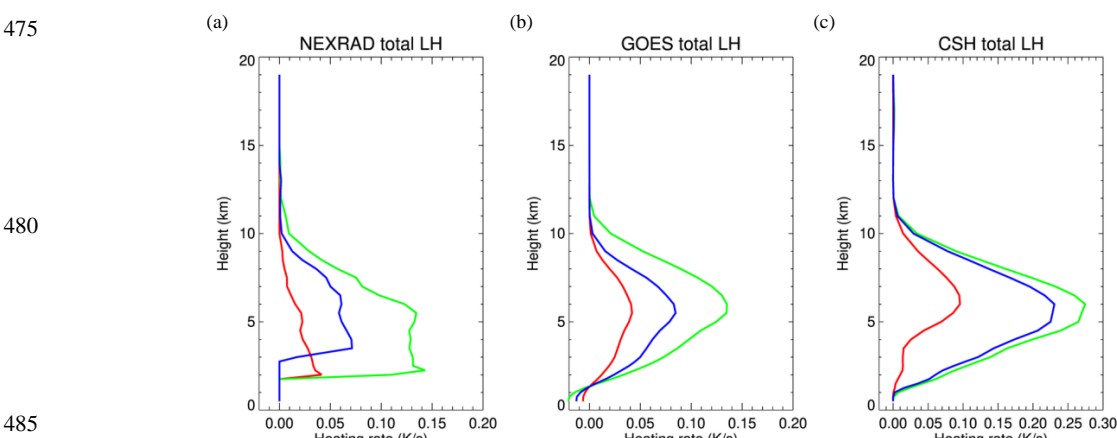

**Figure 7: Vertical profiles of the total heating in the boxed regions from (a) NEXRAD, (b) GOES-16, and (c) CSH. Note that the x-axis of Fig. 7c is different from Figs. 7a and 7b. Different colors represent the color of the box region.**


**Table 4.** Total LH (K/s) from NEXRAD, GOES-16, and CSH in the red, green, and blue box regions.

|  | Red | Green | Blue |
|---|---|---|---|
| NEXRAD | 0.31 | 1.41 | 0.68 |
| GOES-16 | 0.44 | 1.52 | 0.89 |
| CSH | 0.84 | 3.18 | 2.70 |



### 4.2 One-month analysis against NEXRAD LH

A case study from section 4.1 is presented to show how vertical structure of GOES LH differs from other radar products. In this section, one-month of data during June of 2017 are used to compare total LH for convective clouds between GOES-16 and

NEXRAD. Even though the vertical structure is different, its impacts in initiating convection can be similar if the total heating for each convective cloud is similar. Both GOES-16 brightness temperature and NEXRAD reflectivity are resampled to 3km HRRR grid for a direct comparison, and some conditions that are used during convective initiation are applied before the comparison to be useful for the real application. NEXRAD radar reflectivity is converted to LH following Eq. (2) if the layer is cloudy and under GOES cloud top (using Level 2 Cloud Top Pressure data) and if the layer is above the planetary boundary

layer. In case of a layer with temperature greater than 277.15K, reflectivity is only converted to LH if the reflectivity is greater than 28dBZ, while in case of a layer with temperature less than 277.15K, any reflectivity value is converted to LH. These conditions are used during the initialization so that added LH does not disrupt existing model physics too much. Likewise, GOES LH is also set to 0 if the layer is clear or above GOES cloud top, and if the layer is below the planetary boundary layer. Total LH is calculated for each convective cloud from NEXRAD and GOES-16. In the case of NEXRAD, a convective cloud is defined by

combining adjacent grid points whose composite reflectivity is greater than 28dBZ while in the case of GOES-16, adjacent convective grid points by the detection algorithm are clustered to define a convective cloud. In order to minimize errors coming from different definition of convection in GOES and NEXRAD, total LH is compared only in clouds where both NEXRAD and GOES detect convection. Since the area with composite reflectivity exceeding 28dBZ tends to be wider than what GOES-16 defines convective cloud, total LH for each convective cloud system from NEXRAD is compared with total LH from convective

cloud systems from GOES detection that overlap with the convective cloud from NEXRAD. Regions with low radar quality, as indicated by the radar quality flag, are excluded in the analysis.

Using the total of 939 convective clouds collected from the one-month data, the total LH from GOES-16 and NEXRAD are fitted into a linear regression model. Figure 8 shows a scatter plot of NEXRAD LH and GOES LH for each convective cloud,

and the red line is the regression line. A decent correlation coefficient of 0.83 is obtained between NEXRAD LH and GOES LH. According to the slope and the y-intercept of the regression line (1.01 and 1.35 respectively), GOES-16 tends to overestimate LH slightly compared to NEXRAD, which is consistent with the case study result in section 4.1. Around two thirds of the cases show overestimation by GOES, and this is probably because convective clouds with bubbling are detected as convective from GOES but sometimes they do not precipitate, thus undetected by NEXRAD. Most of cases with high discrepancy seems to be caused by

differences in convection detection methods, which is inevitable, but overall, total LH values seem to agree well if the detection was similar.





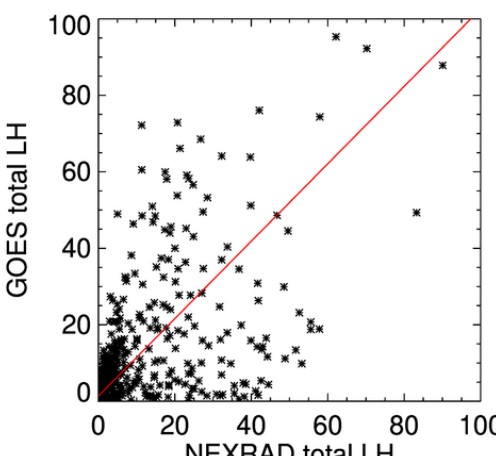


**Figure 8: Scatter plot of NEXRAD total LH and GOES total LH in K/s. Red line is a regression line that is fitted into the one-month dataset.**


## 5 Impacts of NEXRAD LH and GOES LH on precipitation forecast

The WRF model was run for one convective case in 10 July 2019 to compare impacts of GOES LH on precipitation forecast with NEXRAD LH. HRRR data are used as initial and boundary condition, and the same configuration as the one used for making the LUT is used. Results are only compared from 17UTC to 00UTC as GOES-16 visible data are available (15UTC to 22UTC) for
initialization. In order to initiate convection as HRRR does with NEXRAD, observed LH every 15 minute is replaced every time step for 15 minutes with the modeled LH during one hour pre-forecast period. After the pre-forecast run, the model is run freely for an hour, and after the one-hour free run, the one-hour accumulated rainfall rate results are compared. One-hour rain accumulation from simulations without using any observed LH (CTL), using NEXRAD LH (NL), and using GOES LH (GL) are validated against gauge bias corrected quantitative precipitation estimation (QPE; one-hour accumulation) from MRMS.


Figure 9 shows one simulation where observed LH is applied from 15UTC to 16UTC, after which the model is freely run for an hour until 17UTC. The CTL run (Fig. 9a) misses many convective regions, and precipitation is markedly less than MRMS observations in Fig. 9b. Both the NL and GL runs initiated convection in the right place, and enhance precipitation. In the light green box region where CTL run totally misses convection, NL and GL runs both produce precipitation, although there is an
overestimation in NL run while there is an underestimation of precipitation in the GL run. In the dark green box region where convection is weak in the CTL run, both NL and GL runs increased precipitation closer to the observation. The NL run correctly initiates convection in the yellow box region, but not in red box region, while the GL run correctly initiates convection in the red box but not in the yellow box.










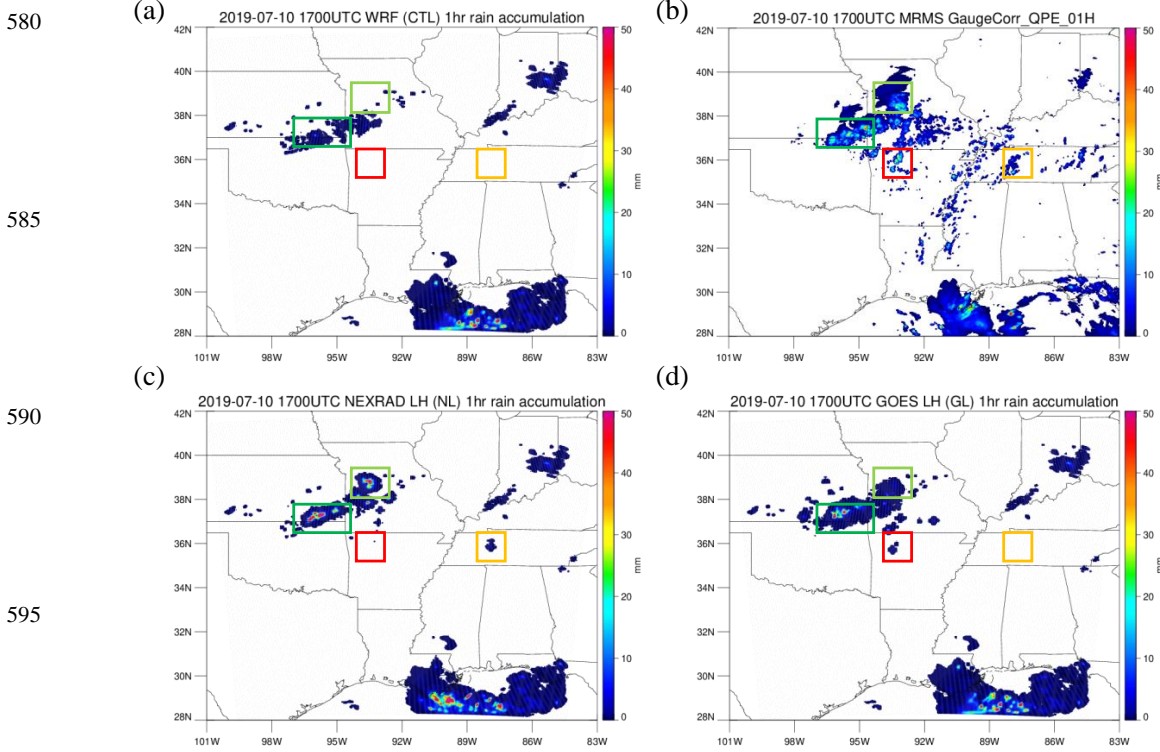


**Figure 9: One-hour rain accumulation at 17UTC in 10 July 2019 from (a) simulation without any LH observation, (b) MRMS gauge corrected quantitative precipitation estimation (QPE), (c) simulation using NEXRAD LH, and (d) simulation using GOES LH.**

These results can be further explained by looking at Fig. 10 which presents maps of vertically integrated NEXRAD LH and GOES LH that are applied to the model at 16UTC which is the last time that observed LH profiles are applied. As seen in the enlarged two green box regions in Fig. 10, NEXRAD shows very high total LH (up to 0.35K/s) in few grid points, and small LH in surrounding area, while most of GOES LH values in the two green boxes are similar and smaller than 0.2K/s. The reason why there was an overestimation in the NL run (Fig. 9c) could be due to this extremely high NEXRAD LH. Interestingly in the red

box region, both NEXRAD and GOES have similar total LH values, but only the GL run produced precipitation (in Fig. 9d). Lastly, it makes sense that GL run did not initiate convection in the yellow box region because no heating is applied due to missed detection (Fig. 10b). Overall, both NEXRAD LH and GOES LH have positive impacts on the precipitation forecast, and their forecast results appear to have similar skills.







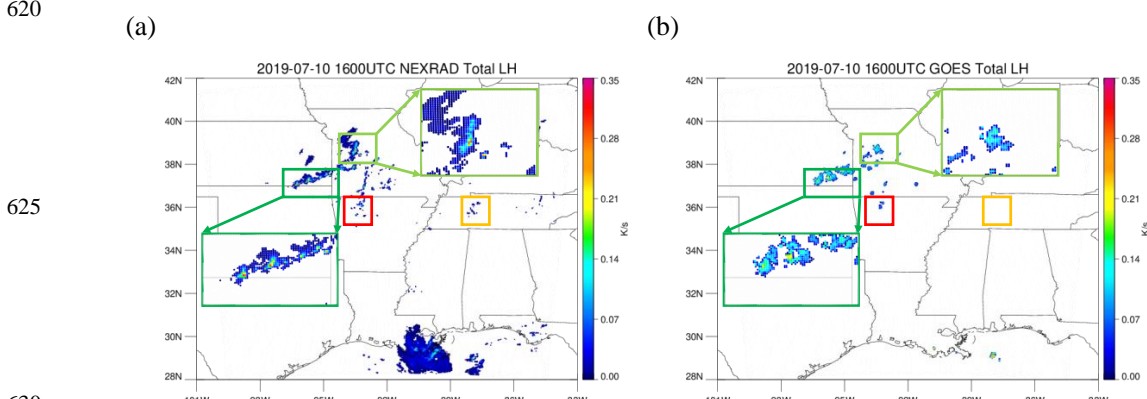

**Figure 10: Vertically integrated LH at 16UTC in July 10ᵗʰ, 2019 from (a) NEXRAD and (b) GOES-16. Two green box regions are enlarged for better comparison.**

For a quantitative evaluation, Fraction Skill Scores (FSS) are calculated for the eight simulations that added LH for different one-hour time periods (LH is added for an hour during 15-16UTC, 16-17UTC, …, 22-23UTC, and FSS are calculated after the one-hour free run at 17UTC, 18UTC, …, 00UTC). FSS is one of the neighborhood-based precipitation verification metrics introduced by Roberts and Lean, 2008, and it is calculated using Eq. (3).

$$FSS_{(n)} = 1 - \frac{\frac{1}{N_x N_y}\sum_{i=1}^{N_x}\sum_{j=1}^{N_y}[O_{i,j} - P_{i,j}]^2}{\frac{1}{N_x N_y}\left[\sum_{i=1}^{N_x}\sum_{j=1}^{N_y}O_{i,j}{}^2 + \sum_{i=1}^{N_x}\sum_{j=1}^{N_y}P_{i,j}{}^2\right]}, \qquad (3)$$

where $N_x$ and $N_y$ are the number of columns and rows, and $O_{i,j}$ and $P_{i,j}$ are respectively an observed and model forecast fraction calculated over a small $n \times n$ domain. It calculates a fraction that passed a threshold value over $n \times n$ domain, and the fraction over the small domain is compared between observation and forecast, rather than comparing the skill at a grid point. In this study, a 15 km × 15 km domain is used to calculate FSS for the six one-hour accumulated precipitation thresholds of 0.254, 2.54, 6.35, 12.7, 25.4, and 50.8 mm/hour (0.01, 0.1, 0.25, 0.5, 1, and 2 inch/hour).


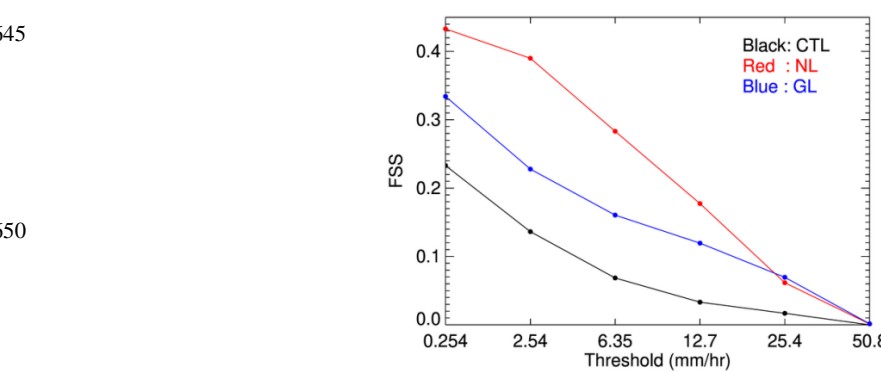



**Figure 11: Fraction Skill Score (FSS) using thresholds of 0.254, 2.54, 6.35, 12.7, 25.4, and 50.8 mm/hour (0.01, 0.1, 0.25, 0.5, 1, and 2 inch/hour) for CTL (black), NL (red), and GL (blue) runs.**





The overall FSS for the eight simulations is shown in Fig. 11. Black, red, and blue lines represent CTL, NL, and GL simulations, respectively. Compared to CTL, both NL and GL runs show significant improvements in FSS for all thresholds. Although NL runs outperform GL at smaller thresholds, GL run shows better results at higher thresholds of 25.4 and 50.8 mm/hour. This is not

surprising given that light rain is more difficult for GOES to detect. Nonetheless, it shows that LH from GOES-16 presented in this study can be useful for improving precipitation forecast especially in the regions where ground-based radar data are not available.

## 6 Conclusions

A method to obtain vertical profiles of LH from GOES-16 ABI data was described. Convective clouds are first detected using temporal changes in reflectance and $T_b$, and LH profiles for the detected cloud are found by searching a LUT created using WRF model simulations. The LUT contains LH profiles of convective clouds that are defined by a threshold of 1.5m/s for the modeled vertical velocity, and these convective LH profiles are sorted according to $T_b$ at 11.2μm, which is a good indicator of cloud top height. Mean profiles that represent each $T_b$ bin show good correlation with cloud top temperature, with lower $T_b$ bin having

deeper LH profiles. Precipitation rates corresponding to each bin are also well correlated to $T_b$. Even though one might think that this method only uses one infrared channel to estimate LH profiles and that is not enough to determine LH intensity, it is actually more than just one brightness value. GOES-16 convection detection algorithm uses 10 time steps of channel 2 reflectance and channel 8 and 10 brightness temperature data to find active convective regions with bubbling cloud top and brightness temperature decrease, and thus the overall algorithm uses more information than just one brightness temperature value. In

addition, LH values in the LUT are well within the range that is allowed in HRRR to initiate convection using NEXRAD, which makes it reasonable to be used to initiate convection in the forecast model.

To investigate how LH from GOES-16 differs from other radar products, LH from GOES-16, NEXRAD, and CSH are compared in three convective clouds with different cloud top heights. Vertical profiles of convective LH from GOES-16 are very similar to

those from CSH that uses model simulations in the LUT. Their vertical profiles show heating throughout the vertical layer except near the surface where evaporation occurs, and heating peaks around the middle of the atmosphere. This vertical pattern differs from when using the empirical formulation used with radar reflectivity by HRRR. Vertical profiles of LH from NEXRAD highly depend on vertical profiles of reflectivity which typically peaks near the surface in convective regions, and thus, maximum LH is usually observed at lower level, which is not commonly shown in the modeled heating rate.


Even though their vertical shape is different, the total LH over convective clouds is shown to be similar. One-month analysis shows that GOES-16 tends to slightly overestimate the total LH for each convective cloud over NEXRAD, but overall good correlation is obtained between GOES-16 and NEXRAD. Furthermore, in order to examine impacts of GOES LH in precipitation forecast compared to NEXRAD LH, one case study is provided. Applying LH derived from GOES-16 was able to

correctly initiate convection in the scene, and the simulation result looks similar to the one applying NEXRAD LH. Although GOES convection detection algorithm is not perfect and misses some convection, and GOES LH is somewhat restricted to cloud top information, these results prove that LH obtained from GOES-16 have reasonable values, and it can be used to improve precipitation forecasts over the region where ground-based radar data are not available.






**Acknowledgments**

This research is supported by the Cooperative Institute for Research in the Atmosphere (CIRA)'s Graduate Student Support Program.

**Author contributions**

All three authors contributed to the retrieval, and the manuscript was written jointly by YL, CK, and MZ.

**Competing interests**

The authors declare that they have no conflicts of interests.

**Data availability**

GOES-16 ABI brightness temperature data are obtained from CIRA, but access to the data is limited to CIRA employees.

GOES-16 ABI Level 2 Cloud Top Pressure (CTP) data are obtained from NOAA National Centers for Environmental Information, Accessed: **[January 25th, 2022]**, doi:10.7289/V5D50K85. GPM DPR data are from: GPM DPR and GMI Combined Convective Stratiform Heating L3 1 month 0.5 degree x 0.5 degree V06, Greenbelt, MD, USA, Goddard Earth Sciences Data and Information Services Center (GES DISC), Accessed: **[January 25th, 2022]**, 10.5067/GPM/DPRGMI/CSH/3B-MONTH/06, GPM DPR Spectral Latent Heating Profiles L3 1 month 0.5 degree x 0.5

degree V06, Greenbelt, MD, USA, Goddard Earth Sciences Data and Information Services Center (GES DISC), Accessed: **[January *25th, 2022*]**, 10.5067/GPM/DPR/SLH/3A-MONTH/06, and GPM DPR and GMI Combined Stratiform Heating L2 1.5 hours 5 km V06, Greenbelt, MD, USA, Goddard Earth Sciences Data and Information Services Center (GES DISC), Accessed: **[January 25th, 2022]**, 10.5067/GPM/DPRGMI/CSH/2H/06. Past MRMS datasets are available at https://mtarchive.geol.iastate.edu/, Accessed: **[January *25th, 2022*]**. HRRR data is obtained from Google Cloud,

https://console.cloud.google.com/marketplace/product/noaa-public/hrrr?project=python-232920&pli=1, Accessed: **[*January 25th, 2022*]**








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
