# Peer review of "Latent heating profiles from GOES-16 and its impacts on precipitation forecasts"

_Atmospheric Measurement Techniques, 2022_

## Author Comment (AC1)

Authors greatly appreciate all the valuable comments and suggestions from the reviewers. Line and figure numbers correspond to the ones in the revised manuscript, and texts that are modified are in red colors in the revised manuscript. A comprehensive read-through is done to correct for English/grammar structure.

Main changes in the revised manuscripts are:
-Figure 7 is removed based on a comment from RC2.
-Table 4 is added.
-Figs. 8 and 9 are modified using different color bars that are colorblind-friendly.
-Fig 10 includes additional run with WSM6 microphysical scheme.
-Section 4.2 is modified using three month of data in 2020.
-Appendix A is added.

RC1

Summary: This study presents a new method for estimating latent heating (LH) profiles from geostationary radiances, and compares the result with established methods that use NEXRAD ground-based radar and TRMM and GPM spaceborne radar. The methodology for estimating LH is similar to what is used for TRMM/GPM LH profiles and is based on a database of output from a convection-permitting resolution model. The authors find that the GOES-based LH estimates are similar to those obtained from NEXRAD and GPM, and produce similar (positive) impact on model forecasts when used in model initialization.

General comments: The use of geostationary data to estimate latent heating is interesting, and potentially valuable, as the Geo data provides much more extensive spatial and temporal coverage relative to NEXRAD and GPM. I think this manuscript is publishable, but needs to be supported with quite a bit more explanation of the tools and datasets used, and also should contain additional context and caveats. I would like the authors to consider the following general recommendations.

1. Any LH estimate from remote sensing is by nature indirect - the observation is of the result of a process that involved LH, not of the LH itself. For example, there can only be hydrometeors for the radar to observe after the condensation process has already happened. A change in the cloud top brightness temperature can only happen after the air has arrived at the top of the storm (having already gone through the condensation process). Please comment on this - I think there is a significant unanswered question that relates to the time and space disconnect between an observation of the result of LH and the LH itself.

A paragraph is added in lines 116-123.

2. Convection is identified using time sequences of GOES imagery, yet the LH profiles are binned by the magnitude of the cloud top brightness temperature. It seems to me that an interesting and

more direct comparison could have been made between the simulated LH and the simulated time-difference brightness temperature. Please discuss.

Authors initially considered time-difference of brightness temperature for the same reason, but decided to bin only with the cloud top brightness temperature as the initial step in this paper for several reasons. Since clouds move over time, calculating change in brightness temperature per pixel can include errors due to cloud advection. In such cases, LH profile had to be assigned per cloud, and we thought that assigning the profiles to individual clouds rather than pixels can make profile inconsistent with the cloud top temperature for each pixel. Another concern related to using time-difference of brightness temperature is in case of mature convective clouds. When clouds reach tropopause, the decrease in temperature is rather small or not observed, and thus, the profiles will look similar anyways. Therefore, it remained as future study. This is discussed in lines 353-365.

3. An obvious point of concern in any model-based lookup table is the model construction and configuration. A 3 km horizontal grid spacing is barely convection permitting, and most simulations of deep convection meant for scientific analysis are now conducted at grid spacings of 1 km or less (most often smaller than 250 meters). Studies comparing simulations with sub-1-km grid spacing with those run at ~3 km have consistently shown that updrafts in 3 km grid spacing simulations are too wide and often too strong, and that the latent heating distribution is shifted higher in the coarse resolution runs relative to the fine resolution runs. In addition, studies have also shown that the LH position and magnitude are very sensitive to the details of the cloud microphysical parameterization. I have a number of questions that I would like the authors to address:

- Why did you not run the WRF model at finer grid spacing? Even if this was not computationally feasible, at least one simulation should be run at fine grid spacing and the LH characteristics compared to assess sensitivity.

The reason for using 3km resolution is to match with spatial resolution of HRRR model. As the reviewer pointed out, the magnitude of latent heating can vary depending on the spatial resolution. Thus, while we understand reviewer's concern in using rather coarser resolution, but the purpose of this study is to use retrieved latent heating to initiate convection at 3km resolution model to remain consistent with the operational HRRR model. In order to keep consistency in magnitude of latent heating between retrieved and modeled ones, we used 3km resolution. The discussion is added in lines 317-320.

- What was the sensitivity of the simulated LH to choice of microphysics? I do not expect a detailed study of this, but as with the previous question, one could imagine running companion simulations of the same case, one with Thompson microphysics and another with (for example) Morrison or WSM6. This would at least provide a first order estimate of the sensitivity.

Authors appreciate the reviewer for raising the great point. Comparing results using different microphysical scheme was part of a future study, but we added one simulation result using WSM6 scheme (Figure 10 in the revised manuscript and line 660-662) to address this point.

4. There was not enough detail provided about the simulation database itself. I was missing the following, which the authors should provide:

More detailed information about the simulation is added in lines 313-320, and Table 2 is modified. It addresses the points below.

- What were the lengths of the simulations (in time)?

It was run for several hours when there was convective activity in the scene.

- What were the geographic domains?

It was mentioned that the geographic domain was over CONUS.

- Which data was used for initial and boundary conditions?

HRRR analysis data are used as initial and boundary conditions

- What was the model output time frequency?

Model is produced every minute, but data every 10 minute are used to create the lookup table.

- How many vertical layers were used? (this can have as large or larger effect on the convection than the horizontal grid spacing)

50 is mentioned in table 2.

- How were the simulations validated? How did the authors ensure they provided a reasonably realistic depiction of storm structure?

It was validated subjectively by comparing simulated brightness temperature and observed brightness temperature.

- Did the simulations span a range of convective types (size, longevity, mode of organization)?

The model was run from the beginning of convective activity to the end over the scene, but the lookup table was not divided into different type of convection as it is hard to distinguish different convective types from observation.

5. The various LH estimates seem to reflect different sources of information on LH. For example, NEXRAD is sensitive to large hydrometeors and primarily obtains information from the lower portions of the troposphere. As such, one would expect the NEXRAD estimates to be biased toward the lower portion of the storm and miss LH in the middle and upper portions. TRMM/GPM radars operate at a shorter wavelength - they will see more of the smaller hydrometeors higher in the storm and may miss some of the heaviest rainfall due to attenuation). One would thusu expect their information to come from the middle portions of the storm but perhaps miss the very lowest and highest layers due to missing detection of heavy rain and small cloud particles. Geostationary data only sees the change in cloud top properties - it's not clear which portion of the storm produces the change at cloud top, but it is likely weighted toward the middle and upper portion of the storm. I would like to see the authors comment on this, and to perhaps discuss how the three sources might be merged in those instances where all three view the same place and time.

Authors agree with the reviewer that there's a potential to merge three products because each observation sees different part of convection. However, the goal of this study is to use LH profiles for short-term forecasts, and DPR product is not suitable for this purpose due to coarse temporal resolution and narrow swath. Yet information from NEXRAD and GOES can be merged through a lookup table in Appendix A which is newly added. Cloud top information from GOES will determine the vertical profile, and the overall intensity can be adjusted using NEXRAD composite reflectivity through the lookup table in Appendix A.

6. There were no caveats listed in the conclusions - one would expect that there are places and times where the GOES data might provide a more reliable estimate of LH and others where these estimates will have larger errors. What are these? Also, there was no mention of future work - what is next? This should also be discussed in the conclusions section.

Lines 690-691 and 700-705 in conclusion are modified to reflect this comment.

Specific comments:

1. June 2017 (the case used to assess impact) is within the time frame used to run the WRF simulations that form the database of profiles. In testing a database-based method, it is common to test on a case that lies outside of the training dataset. I wonder what the results would look like if you compared the estimates for a month from 2019?

We agree that that using June 2017 data is not independent for the testing. We replaced the analysis using summer of 2020. Section 4.2 is modified based on the new analysis.

2. It was clear that there are discrepancies between the NEXRAD and GOES detections of convection. It would be interesting to see statistics on how often these discrepancies occurred.

Statistics of GOES detection accuracy compared to MRMS PrecipFlag product which is different than using 28dBZ but uses NEXRAD radar reflectivity to assign precipitation type is provided in lines 293-294. One third of the three-month data had large discrepancies in detected area (the number of convective grid points from GOES-16 exceeds five times more than the number of convective grid points from NEXRAD and vice versa), and it is added in lines 560-562.

3. The scatter in the plot comparing GOES vs NEXRAD LH in Fig 8 is very large. It is surprising that the correlation was ~0.8. I wonder if the relationship is more obust for smaller LH values than for larger? I suggest using log-log axes for Fig 8 to better be able to examine the smaller LH values.

Thank you for the suggestion. Figure 8 (7 in the revised manuscript) is changed with log-log axes.

4. The phrasing in lines 560-560 on page 16 is confusing - it makes it sound like you are replacing the observed LH with the LH from the model. I think that what you are doing is inserting the observed LH into the model (replacing the modeled LH), right?

Yes, it is rephrased in line 582-583.

5. Follow-up question - are you inserting the observation-estimated LH *profile*? If so, the NEXRAD profile would be bottom heavy while the GOES profile would be top-heavy, right? This would explain the precipitation differences, I would think... NEXRAD LH would produce warming lower in the troposphere, which should result in a much larger effect on buoyancy, relative to GOES.

Yes, the reviewer is correct that we are inserting the vertical profile of LH, and NEXRAD LH would be bottom heavy while GOES would be top heavy. Lines 658-660 are added.

6. While the magnitudes are similar between NEXRAD and GOES estimates of LH, the position of the peak in the vertical matters quite a bit for large scale dynamics. How has this discrepancy been addressed in the literature? Is it assumed that NEXRAD is biased low? Is CSH (and by extension GOES) biased high?

To author's knowledge, there has not been a literature that compares LH from NEXRAD used in HRRR model with LH retrieved from CSH since NEXRAD LH is simply developed to initiate convection in the operational model, and it has not been used to study Impacts of LH in large scale dynamics. Such comparison can be future study.

---

## Author Comment (AC2)

Authors greatly appreciate all the valuable comments and suggestions from the reviewers. Line and figure numbers correspond to the ones in the revised manuscript, and texts that are modified are in red colors in the revised manuscript. A comprehensive read-through is done to correct for English/grammar structure.

Main changes in the revised manuscripts are:
-Figure 7 is removed based on a comment from RC2.
-Table 4 is added.
-Figs. 8 and 9 are modified using different color bars that are colorblind-friendly.
-Fig 10 includes additional run with WSM6 microphysical scheme.
-Section 4.2 is modified using three month of data in 2020.
-Appendix A is added.

RC2

**General Comments:**

Latent heating (LH) is an important process-level cloud variable. To address the temporal gap in LH observations (arising due to satellite estimates only being available periodically), the authors develop a new GOES-based LH retrieval. They compare their GOES LH retrievals with existing satellite and NEXRAD estimates, and then demonstrate the impact (on precipitation forecasts) of assimilating the new LH into WRF. The analyses of impact on precipitation forecasts are fairly minimal (skill scores are shown for a few boxes for a few forecast hours), so this part could be thought of as a proof of concept for using GOES LH in WRF.

My major questions are:

1. Why is only 1 month of data compared across the products? Can additional months be aggregated with some averaging over connected convective features to remove the substantial noise that shows up in instantaneous pixel comparison plots (e.g., Fig. 8)?

   Section 4.2 is modified based on the new analysis, and the plot is modified using 3 months data in 2020.

2. There is a lot of discussion of this quantity: total LH. As best I can tell, this is not an average convective LH, nor is it a vertical integral. It is a sum of convective LH profiles in a box. I did not understand the purpose of just showing the sum, which is largely a result of summing profiles over different convective area counts (and the convective area counts are product dependent, being different for NEXRAD, CSH and GOES). The average profile structure and convective area differences should be portrayed and analyzed separately. If my understanding of total LH computation is incorrect, then it needs to be clarified through the manuscript.

This was done because different techniques have different intrinsic spatial resolutions and thus different magnitudes and profiles. By adding the LH in both the horizontal and vertical axes, we felt that the total heating that is injected into the model could be compared more directly.

3. The analysis of the impact on precipitation forecasts is pretty minimal, and as a reader, I was not convinced that the abstract text stating "improving the forecast significantly" is warranted quite yet. I would re-phrase the text in parts to suggest that this is more of a proof of concept or demonstration of the potential value of assimilating LH.

We agree with the reviewer's point. It is rephrased in lines 26-27, and a sentence to say that it is a proof of concept study is added in the conclusion as well (lines 700-701).

I did not comment on grammar, but I emphasize that a comprehensive read-through is needed to correct many sentences for English/grammar structure. The content overall is appropriate for AMT. My comments above and specific comments below probably warrant a recommendation of major revisions, after which the article -- whose topic is important and intersesting -- will be more useful to the community.

**Specific Comments:**

First line in abstract: It is unusual to say LH is the essential factor driving convective systems. It is also a product of convection. I would reword to say it is an essential factor connected to convective system circulations.

This sentence is rephrased to "affecting intensity or structure of convective systems" in lines 7-8.

Line 30: convection is definitely not resolved explicitly at a few kilometers. Over a decade has passed since the 2011 paper cited, and this is appreciated even more. Perhaps mesoscale convective effects become resolved, but at this resolution, convection is "permitted."

Please see line 31 for an alternate suggestion that simply says that 3km is convection permitting. (This sentence is deleted but "convection permitting" is added in line 30.)

L72: recommend rewriting first part of sentence to: "These products provide instantaneous heating estimates, but their temporal resolutions are low compared…"

It is rephrased to "Although these products have been useful for keeping climate records and understanding impacts of LH in long-lasting systems like tropical cyclones, their temporal resolutions are too low to be used for weather forecasting, especially compared to 2-minute observations available from ground-based radars." in lines 74-76.

L75-80: Can this sentence by written differently? I'm not sure what is trying to be conveyed as written: "Cloud top information from geostationary data is included when creating cloud

analysis during data assimilation (Benjamin et al., 2016), and thus LH retrieved based on cloud top temperature, can be useful in the forecast model by keeping consistency of retrieved LH with the updated cloud analysis."

This sentence is rewritten as "Since the RAP model already uses cloud top information from geostationary data in its forecast (Benjamin et al., 2016), and the HRRR model uses the RAP model outputs as initial and lateral boundary conditions, LH profiles derived from cloud top temperature should be consistent with the model cloud field." in lines 80-82.

L320-322: details about converting units is probably unnecessary information in the article: "but provided in different units. LH from GOES-16 and NEXRAD are in K/s to easily match with modeled heating rate, while DPR products are in K/hour. Therefore, LH in K/hour from DPR products are converted to K/s for comparison."

Authors included these details as other readers of the manuscript had thought that this point was not clear.

L352-353: what is the reason for interpolating to the WRF grid at this point? WRF will rarely get the convection in the exact right place at the right time as observations due to a different surface relative to reality, so why this is done is not clear to me.

Interpolation is done because three products have different resolutions. It doesn't mean that it's interpolated into the WRF grid point where there is convection, but it means that it is interpolated into the same 3km WRF grid. We agree that the sentence is confusing so it is rephrased to "Since the three products have different spatial resolutions, LH profiles from NEXRAD, GOES-16, and CSH for these clouds are interpolated into the same WRF grid with 3km resolution for a direct comparison" in lines 426-428.

L369-370: changes in buoyancy are related to the vertical derivative of heating, not absolute heating rate at any level. If heating is increasing with altitude, then there is a dampening effect on buoyancy specifically. So, it is not about buoyancy here; convection can be initiated because LH increasing with height induces surface convergence which favors convective initiation (which the authors happened to mention near the introduction near L34). Clarify this statement.

Authors agree that the sentence was miswritten. It is rephrased to "This heating at lower levels induces convergence in the lower atmosphere and divergence in the upper atmosphere, and thus, convection can be effectively initiated from the added heating." in lines 444-445.

L374-376: some papers using ARM radars (papers led by Die Wang) indicate that 40 dBZ is a good proxy for convection overall. From that perspective, 28 dBZ is too low too.

We agree that 28dBZ is not a perfect threshold to determine convection, but it is the threshold used in the HRRR model to determine where to apply heating. Therefore, 28dBZ is chosen in this study because the goal of this study is to compare forecast results with what the HRRR model does.

L450-459: I do not understand Fig. 7. For total LH, are you simply summing up all the LH for each convective pixel? So, if for example there are 100 convective pixels, the total LH is the straight sum across all 100 and not an average? Or is total LH some combination of convective for every product + stratiform from CSH? In the literature, total LH as typically reported as the combination of all convective + stratiform + non-raining + anvil LH. If Fig. 7 "total" (and Table 4) is calculated simply by summing convective LH, then I do not understand the value of this figure. It would largely be reflecting differences in convective area and not LH profiles since the previous figures suggested similarity in LH profiles for GOES and CSH. Instead of summing LH, I would strongly recommend showing the convective area differences for each box and product, and then the reader will be able to infer an overall difference in LH as a combination of both convective area differences as well as profile structure differences.

For Figure 7, only the convective LH profiles were used, and horizontally integrated LH was shown. Authors understand the confusion that readers might have by "total LH" used in Figure 7 because we used "total LH" in section 4.2 again but with different meaning. For this reason, authors decided to remove Figure 7, but leave Table 5 that has the same meaning of "total LH" as in section 4.2. Convective regions for each box are shown in Figure 3, but Table 4 is added to show the number of convective grid point so that readers can use Figure 3/Table4 and Figure 6 to guess the results in Table 5.

L510-511: initiating convection certainly depends on structure and not "total" heating. Is there a reference to support that the impact on initiating convection is similar if the total heating is similar? And, again for clarification, what is meant by "total" here – column integrated, or the sum of all individual convective heating profiles?

The "total" in this study is the sum of all individual convective heating profiles, which is horizontally and vertically integrated for each convective cloud. To author's knowledge, there has not been a literature that analyzes impacts of total LH in convective initialization of HRRR model. The authors agree that the horizontal and vertical structures of LH are important factors when it comes to initiating convection. However, GOES LH and NEXRAD LH have different vertical structure as shown in section 4, and convective area detected by GOES and NEXRAD differ because of different detection method. Therefore, it is not reasonable to compare each pixel value at each level, and the only way to provide a good estimate of how the two products differ would be to use total heating. Please also refer to the comment made in "major2". Lines 524-532 are modified to reflect this comment.

L512: the text: "and some conditions that are used…are applied before the comparison to be useful for the real application" is very vague. I do not know what this sentence means.

The whole paragraph (lines 544-550) is modified to explain more concisely.

L529-530: there is a lot of scatter in Fig. 8, and I am uncertain again on what is being shown: the vertical integral ("total") or different levels all combined into a scatterplot of the "total" LH shown for convective regions of Fig. 7? Perhaps the authors should think about doing some averaging of convective LH over the identified convective regions for each rainfall system and then aggregating those convective-system averages and comparing? There is almost too much noise to infer anything from Fig. 8, despite the 0.83 correlation.

What was plotted in Fig. 8 was the total LH of all 939 convective clouds that occurred during June of 2017, not just the convective regions of Fig. 7. This plot was modified to include three months of data in 2020, and the whole section of 4.2 has been changed to include the new analysis. This plot is presented to show that the total heating amounts for each convective cloud from GOES and NEXRAD are within the reasonable range as the case study in section 4.1.

L565-600: Fig. 9, NEXRAD (9c) shows convection in the yellow box, but it disappears when GOES LH was used. Why? I recommend commenting on this. Also – is convective LH assimilated everywhere or only in the 4 boxes?

It is because GOES convection detection algorithm couldn't detect that convection in yellow box, and thus no heating is applied. This was mentioned in the original manuscript, but is now stated more clearly in lines 607-608 to avoid future issues.

L670-672. "Even though one might think… it is actually more than just one brightness value." This sentence is too informal for a publication, and should be re-written.

The sentence is rewritten in lines 671-672.

---

## Author Comment (AC3)

Authors greatly appreciate all the valuable comments and suggestions from the reviewers. Line and figure numbers correspond to the ones in the revised manuscript, and texts that are modified are in red colors in the revised manuscript. A comprehensive read-through is done to correct for English/grammar structure.

Main changes in the revised manuscripts are:
-Figure 7 is removed based on a comment from RC2.
-Table 4 is added.
-Figs. 8 and 9 are modified using different color bars that are colorblind-friendly.
-Fig 10 includes additional run with WSM6 microphysical scheme.
-Section 4.2 is modified using three month of data in 2020.
-Appendix A is added.

RC3
General comments:

Overall, this is an interesting and useful study that has great potential in various applications, because GEO provides much higher spatial and temporal coverages than space-borne and ground-based radars. The proposed method and data product should contribute to convection and precipitation forecasting. For this reason, I'd like to see this paper published.

I only have one major point: the GOE-based LH estimation method described in this study contains two steps: convective cloud detection and LH retrieval for the detected convective cloud. The first step uses a lot more information than the second step. For convective cloud detection, they used multiple channels and their temporal change and spatial structure. In contrast, for LH retrieval, they only used a single piece of information, namely, 11.2-um TB. I wonder if they can consider adding more predictors in their LUT for retrieving LH profiles, given that there are such observations around. Temporal change in TB is an obvious candidate. Meanwhile, environmental parameters will also help. For example, the ambient sounding profile or CAPE has bearings on convective intensity, which should affect the magnitude and vertical structure of LH.

Since the main goal of this study is to use GOES-16 data only to initiate convection, the environmental parameters are not considered. We also decided to bin only with the cloud top brightness temperature for LH estimates in this paper for several reasons.

Since cloud moves over time, calculating change in brightness temperature per pixel can include errors due to cloud advection. In such cases, LH profile had to be assigned per cloud, and we thought that assigning the profiles to individual clouds rather than pixels can make profile inconsistent with the cloud top temperature for each pixel. Another concern related to using time-difference of brightness temperature is in case of mature convective clouds. When clouds reach tropopause, the decrease in temperature is rather small or not observed, and thus, the profiles will look similar. Therefore, it remained as future study.

However, additional lookup table using composite reflectivity is provided in Appendix A. With this lookup table, cloud top information from GOES will determine the vertical profile, and the overall intensity can be adjusted using NEXRAD composite reflectivity through the lookup table in Appendix A. This lookup table can be used with the synthetic radar reflectivity simulator (GREMLIN), but this will be a future study. A paragraph in lines 333-365 is added and lines 700-705 in conclusion section are modified to reflect this comment.

Specific comments:

(Figure 1) The total LH: are they vertical integrals of the LH profile? Can we really integrate LH this way? If LH is 1K/hr at one level and 2K/hr at another level, do we simply add them up? Some clarification is needed.

Vertically integrated LH is used in Figure 1 to show LH in 2-dimensional map. LH at each level is basically a temperature increment at each level, and thus vertically summed value would be temperature increment per hour in one column. In terms of initiating convection, the "total LH" would be the total amounts of LH that will be added to initiate corresponding convective cloud in the forecast model. Definition of "total LH" and the reason for using it is clarified in lines 525-529.

---

## Author Response (AR2)

The authors greatly appreciate reviewers' and editor's time and efforts to reviewing this paper. The whole text is read through once again to correct typos and rephrase a sentence if necessary. Based on the comment, figure 4c,5c, and 6c are changed to include "Conv+Stra mean" and the caption is changed to "The mean of all (convective and stratiform) LH". Corresponding text is also changed in line 429 to "The mean of all LH profile is colored in red.".

---

## Author Response (AR3)

In this round of revision, the manuscript is reviewed by several native English speakers to correct English grammar, and we focused on making sentences flow better. "a", "the", and proper prepositions are modified in many sentences. Only the major sentence changes are listed below with a line number in the revised manuscript.

Line 8: affecting intensity or structure => both the intensity and structure

Line 18: can equally be => can successfully

Line 20: LH profiles from the LUT => LH profiles from a predefined LUT

Line 20: LH from the Next Generation Weather Radar (NEXRAD) => LH used by the HRRR model

Line 31: an effective way to assimilate observation data at this fine resolution has been sought => data assimilation must also be adapted to deal with these finer resolutions

Line 34: convection environment => convective environment

Line 38: LH is not only important to initiate convection, it also contributes to the intensification of convection. => Once the convection is initiated, LH further contributes to the intensification of convection.

Line 45: For this operational purpose => For the operational model

Line 84: with the model cloud field => with both the RAP and HRRR model cloud fields

Line 90: Unlike DPR products that are not available continuously, ABI data in mesoscale sector mode are provided at one-minute resolution, and thus LH can be obtained from GOES-16 as frequently as NEXRAD, making it possible to initiate convection during the forecast. LH from GOES-16 can be beneficial over the regions without radar coverage such as ocean or mountainous regions where beam blockage by terrain degrades the quality of radar data. => In mesoscale sectors of interest, ABI data are provided at one-minute resolution, making the LH product comparable to NEXRAD's product. LH from GOES-16 can be beneficial over the regions without radar coverage such as ocean or mountainous regions where beam blockage degrades the quality of radar data.

Line 94: Detailed descriptions of CSH and SLH products from GPM satellite and how NEXRAD converts reflectivity to LH are provided, followed by the retrieval process using GOES-16 ABI. One case study is provided to compare vertical profiles of LH from GOES-16 with other radar products, and statistical results using three-month of data are provided to evaluate whether total convective heating rates from GOES-16 are comparable to the ones from NEXRAD. Lastly, a Weather Research and Forecasting (WRF) simulation using LH from GOES-16 and NEXRAD is presented to compare impacts of LH from the two datasets in convective initialization. => Detailed descriptions of CSH and SLH products from GPM satellite and how NEXRAD converts

reflectivity to LH are provided in Sect. 2, followed by a description of the LH retrieval from GOES-16 ABI in Sect. 3. Section 4 uses a case study to compare vertical profiles of LH from GOES-16 with other radar products, as well as statistical results over a three-month period to evaluate whether total convective heating rates from GOES-16 are comparable to the ones from NEXRAD. Lastly, in Sect. 5, a Weather Research and Forecasting (WRF) simulation using LH from GOES-16 and NEXRAD is presented to compare impacts of LH assimilation from the two datasets in convective initialization. Results are discussed in Sect. 5.

Line 102: LH is not an easily measurable quantity => LH is not easily measured

Line 105: It is achieved using a diagnostic heat budget method => LH can then be calculated using a diagnostic heat budget method

Line 113: The last six terms on the right-hand side that include these microphysical processes are LH from phase changes. => The last six terms on the right-hand side of Eq. (1) represent the processes responsible for LH.

Line 120: then infer the LH from the hydrometeor content. => then inferring LH from the hydrometeors.

Line 122: However, LH products from ground-based radars, or from a microwave sensor on satellites such as DPR on GPM, can be routinely generated over broad scales, the advantages of which outweigh any time and space mismatch. => Nonetheless, because LH products from ground- or space-based radars and radiometers can be routinely generated over broad scales, the advantages outweigh some of the time and space mismatches.

Line 132: briefly explained here => briefly explained here for completeness

Line 135: The initial algorithm by Tao et al.1993 used surface rainfall rate and amount of stratiform rain as inputs to the LUT, but the LUT has been improved by increasing the number of LH profiles, using finer resolution in simulations, and adding new inputs such as echo-top heights and low-level vertical reflectivity gradients => The initial algorithm by Tao et al.1993 used surface rainfall rate and amount of stratiform rain as inputs to a LUT that was generated from a number of representative cloud model simulations. This LUT has since been improved by increasing the number of simulations, using finer resolution in simulations, and adding new variables such as echo-top heights and low-level vertical reflectivity gradients

Line 141: "to select the appropriate LH profile" is added.

Line 144: "the work of" is added.

Line 144: the LUT is created for three different rain types => the LUT is created from cloud resolving model simulations for three different rain types

Line 151: For DPR, a new LUT is created for => The DPR uses a new LUT created for

Line 152: For higher latitude regions, six precipitation types (convective, shallow stratiform, three types of deep stratiform, and other) are used instead of three, and therefore six respective LUTs exist. Inputs to these LUTs are precipitation type, PTH, precipitation bottom height, maximum precipitation, and $P_s$. => Cloud in higher latitude regions is classified into six precipitation types (convective, shallow stratiform, three types of deep stratiform, and other). This creates six LUTs that provide LH as a function of precipitation type, PTH, precipitation bottom height, maximum precipitation, and $P_s$.

Line 158: "distributions" is added.

Line 162: comprise => generate

Line 234: Orbital data for these products have finer spatial resolution of 5km => Orbital data for these products is provided at the pixel scale (5km)

Line 235: "from low Earth orbit" is added.

Line 239: in the right places => at the appropriate locations

Line 239: obtained through => constructed using

Line 256: computational instability => computational instabilities

Line 259: moved around to observe interesting => selected around to observe important

Line 270: on available variables => on the variables available

Line 280: several values are tested to produce corresponding convective fractions. => in order to match the convective fraction seen in the GOES-16 convection detection algorithm (described in Lee et al., 2021).

Line 286: measures $T_b$ decrease => focuses on $T_b$ decreases

Line 291: one-month analysis against => one-month of data compared to

Line 300: Using 1.5m/s => Using a 1.5m/s threshold

Line 307: in observation => from observations

Line 311: or => and

Line 313: "any" is added.

Line 322: gathered => included

Line 327: are collected according to 16 bins of the cloud top temperature at 11.2 µm. => are sorted into 16 bins based on the cloud top temperature at 11.2 µm.

Line 330: It is also nicely shown in the figure that as the $T_b$ decreases => It is also clear in the figure that as the $T_b$s decreases

Line 356: but in such a case, LH profiles will have to be assigned for each cloud, and the assigned profile will be inconsistent with the observed cloud top temperature for each pixel. => However, LH profiles would have to be assigned for each cloud, and the assigned profile would be inconsistent with the observed cloud top temperature for each pixel.

Line 359: it remains as future study => it remains for future studies

Line 361: whole => full

Line 368: "in this section" is added.

Line 376: to compare how each product determines precipitation type (convective or stratiform) which is one of the major factors in estimating LH profiles => to compare the precipitation types (convective or stratiform) of the three products, as this is one of the major factors in estimating LH profiles

Line 380: the smallest regions compared to others => convective areas relative to the other two methods

Line 437: On the other hand => In contrast

Line 444: Overall values of mean LH profile from NEXRAD in blue are slightly smaller than mean profile of GOES LH or mean convective LH profile from CSH (blue line), but are closer to the total mean profile of CSH (red line), which indicates that the 28dBZ threshold might include some stratiform regions as well. => Overall values of the mean convective LH profiles from NEXRAD in blue are slightly smaller than the mean convective profile of GOES LH and CSH (blue line), but are closer to the total mean profile of CSH (red line), which indicates that the 28dBZ threshold might include some stratiform regions as well.

Line 522: added => summed

Line 523: than GOES-16 detection in Fig. 3b => relative to that of GOES-16 (Fig. 3b)

Line 525: The reason why the total LH is used for a comparison is because NEXRAD LH has such a different vertical structure from GOES LH or CSH LH and such different convective areas, that it is difficult to makes direct comparison between vertical levels. => This comparison is intended

to account for differences in the area and convective definitions that make direct comparison between vertical levels difficult.

Line 542: horizontally integrated LH over => horizontally integrated over

Line 558: from the three-month data => during the three-month of the analysis

Line 562: in log-log axes => using log-log space

Line 563: high discrepancy => large discrepancies

Line 581: as HRRR does with NEXRAD => in the same manner as HRRR does with NEXRAD

Line 594: precipitation closer to the observation => precipitation amounts closer to the observations

Line 683: highly depend on => strongly depend on

Line 683: and thus, maximum LH is usually observed at lower level, which is not commonly shown in the modeled heating rate. => This leads the NEXRAD maximum LH to be at lower levels, not often simulated in the models.

Line 693: Applying LH derived from GOES-16 => Applying LH derived from GOES-16 to model initialization

Line 701: additional wind products will be needed for both model and observation to remove errors coming from cloud advection => additional wind products will be needed to remove model and observational errors coming from cloud advection

Line 703: In addition, more investigation will be needed => Further investigation will also be needed